# Structure-Based Discovery of Receptor Activator of Nuclear Factor-κB Ligand (RANKL)-Induced Osteoclastogenesis Inhibitors

**DOI:** 10.3390/ijms241411290

**Published:** 2023-07-10

**Authors:** Vagelis Rinotas, Fotini Liepouri, Maria-Dimitra Ouzouni, Niki Chalkidi, Christos Papaneophytou, Mariza Lampropoulou, Veroniki P. Vidali, George Kontopidis, Elias Couladouros, Elias Eliopoulos, Athanasios Papakyriakou, Eleni Douni

**Affiliations:** 1Institute for Bioinnovation, Biomedical Sciences Research Center “Alexander Fleming”, 34 Fleming Street, 16672 Vari, Greece; 2proACTINA SA, 20 Delfon Street, 15125 Athens, Greece; 3Laboratory of General Chemistry, Department of Food Science and Human Nutrition, Agricultural University of Athens, 75 Iera Odos, 11855 Athens, Greece; 4Department of Biochemistry, Veterinary School, University of Thessaly, 224 Trikalon, 43131 Karditsa, Greece; 5Department of Life Sciences, School of Life and Health Sciences, University of Nicosia, 46 Makedonitissas Avenue, 2417 Nicosia, Cyprus; 6Institute of Nanoscience and Nanotechnology, National Centre for Scientific Research “Demokritos”, Patr. Gregoriou E & 27 Neapoleos Str, 15341 Athens, Greece; 7Laboratory of Genetics, Department of Biotechnology, Agricultural University of Athens, 75 Iera Odos, 11855 Athens, Greece; 8Institute of Biosciences and Applications, National Centre for Scientific Research “Demokritos”, Patr. Gregoriou E & 27 Neapoleos Str, 15341 Athens, Greece

**Keywords:** virtual screening, computer-aided drug discovery, small-molecule inhibitor, cell-based assay, toxicity evaluation, synthesis, compound solubility

## Abstract

Receptor activator of nuclear factor-κB ligand (RANKL) has been actively pursued as a therapeutic target for osteoporosis, given that RANKL is the master mediator of bone resorption as it promotes osteoclast differentiation, activity and survival. We employed a structure-based virtual screening approach comprising two stages of experimental evaluation and identified 11 commercially available compounds that displayed dose-dependent inhibition of osteoclastogenesis. Their inhibitory effects were quantified through TRAP activity at the low micromolar range (IC_50_ < 5 μΜ), but more importantly, 3 compounds displayed very low toxicity (LC_50_ > 100 μΜ). We also assessed the potential of an *N*-(1-aryl-1*H*-indol-5-yl)aryl-sulfonamide scaffold that was based on the structure of a hit compound, through synthesis of 30 derivatives. Their evaluation revealed 4 additional hits that inhibited osteoclastogenesis at low micromolar concentrations; however, cellular toxicity concerns preclude their further development. Taken together with the structure–activity relationships provided by the hit compounds, our study revealed potent inhibitors of RANKL-induced osteoclastogenesis of high therapeutic index, which bear diverse scaffolds that can be employed in hit-to-lead optimization for the development of therapeutics against osteolytic diseases.

## 1. Introduction

Bone homeostasis is regulated by a balanced process known as bone remodeling, which functions through a constant interplay between bone resorption and bone formation for the replacement of old or damaged bone. Osteoblast cells are responsible for bone formation through the synthesis and secretion of bone matrix proteins such as collagen type I, and eventually they mineralize the bone matrix through deposition of hydroxyapatite crystals [1]. On the other hand, osteoclasts are multinuclear giant cells responsible for bone resorption through acidification of the microenvironment, which dissolves the mineralized bone matrix, and secretion of enzymes like cathepsin K and tartrate-resistant acid phosphatase (TRAP) that degrade the organic components [2]. Impairment of bone remodeling can lead to various skeletal disorders, including postmenopausal osteoporosis, which is characterized by low mineral density and increased risk of fractures due to an increased bone remodeling rate and osteoclast activity [3]. Receptor activator of nuclear factor-κB ligand (RANKL), a tumor necrosis factor (TNF) superfamily member, constitutes the master regulator of bone resorption. RANKL, existing both as a type II transmembrane protein and as a soluble form, binds to its cognate receptor RANK, activating a signaling cascade of events essential for the differentiation, activation and survival of osteoclasts [4,5]. RANKL is expressed also in other cell types, such as activated T lymphocytes, fibroblasts, synoviocytes and mammary epithelial cells, and thus it has been implicated in diverse in vivo biological processes including immune regulation, mammary gland development, hormone-induced breast cancer and body thermoregulation [6,7,8]. Genetic deletion of either RANKL or RANK results in autosomal recessive osteopetrosis, a rare genetic disease characterized by osteoclast absence or malfunction [9,10,11]. A decoy soluble receptor of RANKL known as osteoprotegerin (OPG) prevents the binding of RANKL to RANK, and thus protects against aberrant osteoclastogenesis and bone resorption [12,13,14].

Denosumab, a monoclonal antibody that binds to human RANKL and prevents its binding to RANK receptor [15], specifically inhibits the activity of osteoclasts and has been approved for the treatment of postmenopausal osteoporosis since 2010 [16,17]. Apart from Denosumab, there are alternative anti-resorptive therapies including bisphosphonates that lead to a reduction in osteoclast activity, as well as osteo-anabolic therapies stimulating bone formation [18,19]. However, despite the effectiveness of the established anti-osteoporotic therapies, numerous concerns have arisen regarding their safety and efficacy. For instance, bisphosphonates have been implicated in osteonecrosis of the jaw after long-term exposure, while the osteo-anabolic therapies are restricted to a maximum of two years of therapy since there are concerns about tumorigenesis. In addition, the use of antibodies in therapies has been correlated with inadequate pharmacokinetics, low tissue accessibility, and increased immunogenicity and relapse of osteoporosis after discontinuation [20,21]. Considering these functional limitations, further research is required in the context of drug discovery targeting bone resorption through alternative therapeutic approaches.

On these grounds, several studies have investigated the use of peptidomimetics [22,23] and compounds of small molecules (compounds of MW < 900 Da) [24,25,26,27,28,29,30,31] that inhibit activity of RANKL. Early efforts have focused on the structure-based design of peptide segments from loops of OPG, a natural inhibitor of RANKL with a binding affinity similar to that of RANK receptor [22,23], and the discovery of small molecules that inhibit RANKL-induced osteoclastogenesis. These inhibitors were identified mainly through high-throughput screening, with representative compounds from benzopyranyl tetracycles (**7ai**) [24], indeno [1,2-*c*]quinolones (**8a**) [25] and pyridinylpyrimidine derivatives (**AS2676293**) [26], as shown in Figure 1. Others have focused on modified salicylanilides (**6i**, Figure 1) [28] and salicylanilide-derived small molecules such as 2*H*-benzo[*e*][1,3]oxazine-2,4(3*H*)-diones (**5d**, Figure 1) [27], based on the inhibitory effect of sodium salicylate in bone resorption [32].

We recently presented a ligand-based approach that revealed several analogues of SPD-304, a small-molecule inhibitor of TNF-α [33], as potent inhibitors of RANKL-induced osteoclastogenesis with low toxicity (e.g., **19b, Figure 1**) [29]. Other recent works were based on hit compounds that were identified through structure-based molecular docking of commercially available compounds (the SPECS database of >200,000 small molecules) [34], or developed lead compounds such as the β-carboline derivative **Y1693** (Figure 1) [31]. Here, we present the identification of 11 potent inhibitors of RANKL-induced osteoclastogenesis (IC_50_ < 5 μΜ) through a two-stage approach comprising virtual screening of three commercially available small-molecule databases and a similarity search. We also present the synthesis of 30 analogues of a designed scaffold that was based on one of the initial hits, from among which 4 of the synthetic compounds are potent inhibitors. These 15 inhibitors of osteoclastogenesis comprise five distinct scaffolds, one of which exhibited both a high hit rate and low-toxicity derivatives. Structure–activity relationships in models of the hit compounds bound to a human RANKL dimer provide key residue-specific interactions at the targeted pocket, which could be employed in future hit-to-lead optimization efforts.

## 2. Results and Discussion

### 2.1. Structure-Based Discovery of the Inhibitors

With the aim to identify new scaffolds that could be employed as potent inhibitors of RANKL-induced osteoclastogenesis, we targeted the interface of a human RANKL dimer model with small molecules from libraries of commercially available compounds. This hypothesis was based on our previous finding that SPD-304, a small-molecule inhibitor of TNF trimer formation [33], also inhibits RANKL-induced osteoclastogenesis in a dose-dependent manner [29]. Using chemical cross-linking of soluble human RANKL with SPD-304, we demonstrated that the inhibitor mediates dissociation of the biologically active trimers. Therefore, based on the X-ray structure of an intermediate TNF dimer in complex with SPD-304 (PDB ID: 2AZ5) [33], we determined the corresponding site on a model of the RANKL dimer. This site was employed in virtual screening of more than 100,000 compounds that were compiled from three commercially available libraries. Although the total number of compounds was orders of magnitude lower than the current state-of-the-art in virtual screening (hundreds of millions to some billions of compounds) [35,36,37], we took a chance based on the diversity of the libraries employed and the computational power available at that time. For this reason, we selected the MyriaScreen Diversity Collection from Aldrich (10,000 compounds), the DIVERSet library from ChemBridge (50,000 compounds) and the BIONET Screening compounds from Key Organics (42,839 compounds). Some basic compound chemical properties of the three libraries are shown in Appendix A).

The docking scores obtained using AutoDock VINA [38] were used primarily to identify the top-ranked compounds within a range of 2–3 kcal/mol (Appendix A). Approximately 1% of the top-ranked compounds from each library were visually investigated for hydrogen bonding interactions, and hydrophobic and aromatic contacts, as well as the overall shape complementarity at the docking site of the RANKL dimer. We paid much attention to this time-consuming step (a total of more than 1000 compounds were inspected), since human intervention has been shown to improve the prediction performance of virtual screening in most cases [39]. In this way, we selected a subset of possible inhibitors (~150), from which 10 compounds from each library were finally cherry-picked (Figure 2 and Appendix A). Their selection was primarily based on their diversity and predicted interactions with key residues of RANKL (see below), without additional filters for drug or lead-likeness being applied at this stage [40].

### 2.2. Evaluation of the Compounds in Osteoclastogenesis Assays

A key property of the compounds used for screening was solubility, which was assessed as described previously [41]. Herein, we report solubility as observed in 5 mM stock solutions in 100% DMSO, and upon dilutions to 0.3 mM in phosphate-buffered saline (PBS) pH 7.4 containing 5% DMSO. Of the 30 compounds obtained, 7 displayed low solubility even at 100% DMSO and were discarded without further evaluation (Low solubility, Table 1). There were 6 compounds that were soluble at stock solutions but displayed limited precipitation at 5% DMSO in PBS [41] and were thus screened at tentative concentrations (Medium solubility, Table 1). Even so, these 6 compounds proved to be inactive in mediating RANKL-induced osteoclastogenesis. The remaining 17 compounds were readily soluble and did not display any aggregation upon gradual dilution in aqueous media (High solubility, Table 1). A comparison of their calculated logarithm of the partition coefficient in n-octanol/water mixture (cLogP, Table 1) revealed a very poor correlation with the observed solubility. Although most of the low-solubility compounds had cLogP values greater than 4.0, we identified 13 highly soluble compounds with cLogP > 4.0 (with values as high as cLogP = 5.9), which would have been excluded as non-lead-like compounds. Irrespective of the poor correlation between the cLogP and the observed solubility, we did not apply any filtering criteria for size or hydrophobicity, driven by the fact that most potent protein–protein inhibitors are of high molecular weight, hydrophobicity and aromaticity [42].

The first set of 23 soluble compounds were evaluated for their inhibitory effects stimulating either bone marrow (BM) cells or the macrophage cell line RAW264.7 in a RANKL-induced osteoclastogenesis assay with a concentration cutoff of 5 μΜ. Our results showed that 15 compounds did not exhibit any inhibitory effect, and 4 compounds displayed partial inhibition only; thus, these compounds failed to pass the criteria for further investigation (Table 1). However, 4 compounds showed total inhibition of RANKL-induced osteoclast formation at 5 μΜ (Figure 1A), and these were further evaluated to determine their half-maximal inhibitory concentration (Figure 1B) and cellular toxicity (Figure 1C). Compound **5J-319S** displayed the lowest value IC_50_ in TRAP activity (1.1 μM) with moderate cellular toxicity (LC_50_ = 41.5 μΜ), whereas **7756003** showed moderate activity (IC_50_ = 4.6 μM) with higher toxicity (LC_50_ = 33.5 μΜ). Still, the observed inhibition of osteoclast formation was not associated with any cellular toxicity at the ranges examined (Figure 1). Interestingly, compounds **8P-504S** and **6747072** displayed very low cellular toxicities, exhibiting LC_50_ values of over 100 and 200 μM, respectively, while retaining a high inhibitory effect on RANKL-induced osteoclastogenesis (Table 1). It should be noted that hit compounds displayed a similar range of inhibitory and toxicity effect, irrespective of the cell type employed, as indicated by comparing the IC_50_ and LC_50_ obtained in the macrophage cell line RAW264.7 compared to those in BM cells (Appendix A). Taken together, our initial screen identified 4 hit compounds that inhibited RANKL-induced osteoclast differentiation with low cellular toxicity.

### 2.3. Structure–Activity Relationships of the Hit Compounds

Based on the results obtained, we identified four diverse hit compounds as potent inhibitors of RANKL-induced osteoclastogenesis, two of which displayed low cell toxicities (LC_50_ > 100 μΜ) as well. Key interacting residues of RANKL at the targeted site comprised the aromatic Tyr215, Tyr217 and Phe311, and the polar Asn276, with commonly observed hydrogen-bonding interactions with the phenolic groups of Tyr215 and Tyr307, and the amide NH_2_ of Asn276, as donors (Figure 2). Interestingly, three out of the four hits contained a sulfonamide moiety that interacts with Asn276 (Figure 2B,D,E), similar to several potent inhibitors identified in our previous work [29]. Although another sulfonamide-containing compound (R897698, Figure 2) was not tested due to low solubility, this observation prompted us to adopt a sulfonamide-containing scaffold for a series of compounds that were synthesized and screened as RANKL inhibitors (see below). The fourth hit compound (**7756003**) did not contain sulfonamide but a thioether; however, the adjacent carbonyl group was predicted to interact with Asn276 (chain A) as a hydrogen bond acceptor (Figure 2C), and an additional hydrogen bond could be formed with the side chain phenol of Tyr307(A). Similarly, **6747072** displayed hydrogen bonds with the phenolic groups of Tyr307(A) and Tyr215(B) (Figure 2B), whereas **8P-504S** showed the potential to accept a hydrogen bond from the phenolic oxygen of Tyr215(A) (Figure 2D). Other polar contacts included the interaction between the trifluoromethyl group of **5J-319S** with the phenolic oxygen of Tyr215(A) and the carbonyl group of Asn276(B) (Figure 2E). Taken together, we can conclude that the four hit compounds exhibited the potential to form a combination of aromatic π–π interactions with key aromatic residues of the binding pocket, while placing hydrogen-bond acceptors close to the exposed phenolic and amidic side chain groups of Tyr215, Tyr217, Tyr307 and Asn276.

### 2.4. Hit-Based Discovery of Potent Inhibitors

With the aim to investigate the potential of these four scaffolds for further improvement, we employed a similarity search for closely related analogues within the ZINC12 database [43], in addition to a synthetic approach using a scaffold similar to that of the hit compound **5J-319S** (see below). From the similarity search within the purchasable space of compound **6747072**, we found only a single derivative, whereas for compounds **7756003**, **5J-319S** and **8P-504S** several analogues were retrieved, from which we selected seven, nine and four compounds for evaluation, respectively (Figure 3 and Appendix A).

The high solubility of the initial hits was a key property of the compounds, so the selected 21 analogues displayed good solubility in 100% DMSO, with only 4 compounds showing some precipitation upon dilution in 5% DMSO/phosphate-buffered saline (Medium solubility, Table 2). Evaluation of the second set of 21 compounds revealed seven additional hits, from among which **7774021** displayed the lowest toxicity in BM cells (LC_50_ > 200 μΜ, Figure 3). From the single derivative of **6747072** and the analogues of **8P-504S**, we did not observe any inhibition of osteoclastogenesis (Table 2). This result is very challenging to interpret on a structural basis, given that compound 7,553,178 bears only an additional methyl group compared to **6747072**, whereas the p-methyl group of **8P-504S** is substituted by a halogen in 8P-505S and 8P-517S (Figure 3).

Among the nine selected analogues of **5J-319S**, two compounds displayed total inhibition of osteoclastogenesis, **5J-359S** and **6J-323S**, albeit with lower activity than the parent compound (Table 2 and Appendix A). The latter exhibited lower toxicity compared to the parent compound, although this result should be considered carefully due to the partial solubility of **6J-323S** observed upon dilution in 5% DMSO/phosphate-buffered saline. Despite that, the dimethyl-1*H*-benzimidazole scaffold of this series was suggested as a putative starting point for lead optimization. Interestingly, most hits were analogues of **7756003** with improved activity (IC_50_ < 3.3 μΜ), and two of them also displayed significantly reduced toxicity (**7753688** and **7774021**, Figure 3). Their lower toxicity was probably due to substitution of the methylenethio-trimethyl-pyrimidine moiety of the parent hit by cyclohexyl or p-chloro-phenyl groups, respectively, although replacement of the carbonylic moiety by pyridine in **7747909** resulted in higher toxicity (Table 2). The remaining three hit compounds supported the observation that 1-(piperazin-1-yl)-4-(*p*-tolyl)phthalazine is a very promising scaffold for development of potent RANKL inhibitors that display low cellular toxicity. Regarding the two inactive analogues of **7756003** (7757551 and 7771348, Table 2), their medium solubility could be one reason for their failure to inhibit osteoclastogenesis at 5 μΜ. However, several other factors could be in play too (e.g., cellular permeability, low affinity for RANKL), and docking has an intrinsically high false positive rate. Therefore, we were not able to provide meaningful structural information regarding their inactivity, especially considering the lack of experimental structures of RANKL in complex with small-molecule inhibitors.

Comparison of the predicted bound poses of the hit compounds revealed that the orientation of the 1-(piperazin-1-yl)-4-(*p*-tolyl)phthalazine moiety of **7756003** and of its five analogues (Figure 4A–E) was very similar, with the exception of **7747909** that lacks the *N*-linked carbonyl substituent of piperazine (Figure 4E). At this orientation, the pyridine substituent in **7747909** could accept a hydrogen bond from Asn276, whereas the carbonylic substituent in the other analogues may either interact with monomer A (**7756003**, **7775352**, **7753688**) or monomer B (**7774021**, **7775390**) of the RANKL dimer. It should be noted, however, that although regular aromatic interactions with Tyr215, Tyr217 and Phe311 were observed, hydrogen bonding interaction with Asn276 was predicted only for **7747909**. The amidic carbonyl of the other hits can accept a hydrogen bond either from Tyr215(B) (**7774021**, **7775390**, Figure 4B,C) or Tyr217(A) (**7775352**, **7753688**, Figure 4C,D). Similarly, the two hit-analogues of **5J-319S** displayed diverse docked poses, with the sulfonamide group of **5J-359S** interacting with Asn276(A) (Figure 4G) and the sulfonamide group of **6J-323S** with both Tyr215(A) and Tyr215(B) (Figure 4H). One of the two methoxy groups of **5J-359S** may also accept hydrogen bonds from the NH_2_ group of Ans276(B) and the main chain NH of Gly278(B), whereas the tert-butyl substituent of **6J-323S** exhibited hydrophobic contacts with Val313 (Figure 4G,H). Taken together, these observations indicated that variable substituents in the hit compounds of their analogues will probably mediate diverse bound poses; still, residue-specific interactions of their polar groups should account for their activity. Experimental data from X-ray crystal structures is still necessary in order to obtain more detailed structure–activity relationships and guide hit-to-lead optimization.

### 2.5. Design and Synthesis of PRAN Compounds

Although the 1-(piperazin-1-yl)-4-(*p*-tolyl)phthalazine scaffold of **7756003** was present in five additional hits, we designed a scaffold based on **5J-319S** and the two hit-analogues thereof (Figure 4, hereafter PRAN compounds). For its design, we considered (i) the presence of a sulfonamide moiety in three diverse scaffolds, including hits **6747072** and **8P-504S**; (ii) the occurrence of an indole ring in several potent inhibitors of RANKL (Figure 1); and (iii) the ease of synthesis (efficiency and step economy). The benzylic substituents in PRAN were replaced by aryl groups in analogy to our previous work that was based on SPD-304 (e.g., **19b** in Figure 1). Therefore, synthesis of compounds PRAN-1.1 to PRAN-3.10 was carried out starting from a Cu(I)-catalyzed coupling of aryl bromide **2** with 5-nitro-1*H*-indole (**1**), platinum-catalyzed reduction of the nitro-product **3** to the corresponding amine **4**, and coupling of the amine with the desired sulfonyl chloride **5** (Figure 5 and Appendix A).

Evaluation of the 30 synthetic PRAN compounds revealed an overall high solubility in DMSO, even for the mono- and disubstituted trifluoromethlphenyl compounds PRAN-x.9–x.10 (where x = 1, 2, 3), for which a relatively high cLogP was estimated (Table 3). With regard to their activity, 4 compounds displayed total inhibition of osteoclastogenesis at 5 μΜ and two derivatives showed partial inhibition at the same concentration (Table 3 and Appendix A). The inhibitory effect was quantified with TRAP staining and revealed IC_50_ values in the range of 2.0–4.3 μΜ; however, their toxicity as evaluated in BM cells exhibited LC_50_ values of 11–24 μM. Due to the low therapeutic index of the PRAN compounds, we did not make any effort to extract structure–activity relationships, and although we evaluated a limited set of compounds (those that showed total inhibition of osteoclastogenesis at 5 μΜ), our results suggested that the aryl-substituted *N*-(1*H*-indol-5-yl)sulfonamide scaffold may not be appropriate due to undesirable toxicity considerations.

## 3. Conclusions

Through structure-based virtual screening of commercially available compounds, we identified four compounds that showed dose-dependent inhibition of RANKL-induced osteoclastogenesis, with low micromolar inhibitory effects (IC_50_ < 5 μΜ) as quantified by TRAP staining. From the most potent hit (**5J-319S**, IC_50_ = 1.1 μΜ), we designed a *N*-(1-aryl-1*H*-indol-5-yl)aryl-sulfonamide scaffold and assessed its potential through the synthesis of 30 derivatives (Figure 4 and Figure 5). Although 4 of these derivatives displayed total inhibition of osteoclastogenesis at low micromolar concentrations, toxicity concerns hamper their potential for further development. We also evaluated nine commercially available analogues of **5J-319S** (Figure 3) and discovered two additional hits with comparable therapeutic potential (Table 2). For another hit compound that displayed the lowest toxicity, **8P-504S** (LC_50_ < 200 μΜ), we also evaluated four analogues of high similarity that did not display inhibition of osteoclastogenesis in cells. This was also the case with the initial hit compound **6747072** (IC_50_ = 2.9 and LC_50_ > 100 μΜ), for which we found only a single methylated analogue with no activity (Figure 3). On the other hand, the 1-(piperazin-1-yl)-4-(*p*-tolyl)phthalazine scaffold of the initial hit compound **7756003** proved to be very promising, as supported by the identification of five additional hits out of seven commercially available analogues that were evaluated (Table 2). Importantly, these series provided not only the highest hit rates, but also several low toxicity compounds. Their structure–activity relationships using models of their bound complexes suggested key interacting residues of human RANKL dimer that could be targeted specifically. It has to be noted, however, that to the best of our knowledge there is no crystallographic structure of the RANKL dimer bound to a small-molecule inhibitor of RANKL trimer formation; and, although we targeted a specific site on a model of human RANKL dimer using virtual screening, we cannot rule out the possibility that the observed RANKL-induced osteoclastogenesis was due to inhibition of RANKL binding to RANK. Further crystallographic efforts with RANKL-induced osteoclastogenesis inhibitors and RANKL are warranted. Taken together, our study revealed potent inhibitors of RANKL-induced osteoclastogenesis from diverse scaffolds that can be employed in hit-to-lead optimization for the development of therapeutics against osteolytic diseases.

## 4. Methods and Materials

### 4.1. Computational Methods

The structure of human RANKL dimer employed in virtual screening was obtained as described in our latest work [29]. Briefly, a single monomer of RANKL was extracted from the asymmetric unit of the X-ray structure of RANKL trimer in complex with the N-terminal fragment of its decoy receptor osteoprotegerin (PDB ID: 3URF) [13] and then superimposed with each of the two chains in the X-ray structure of TNF in complex with the small molecule SPD-304 (PDB ID: 2AZ5) [33]. In this way, we prepared a model of a human RANKL dimer with the two subunits slightly widened with respect to the native trimer, which is a more suitable target for small molecules. AutoDockTools v.1.5.4 [44] was used to prepare the protein for docking and assign the search space at the center of the targeted pocket with dimensions of 25 × 25 × 20 Å. A single, low-energy conformation for each compound was calculated from the SMILES representations provided by each vendor, using OMEGA v.2.3 (OpenEye Scientific Software, Santa Fe, NM, http://www.eyesopen.com, accessed on 21 June 2023) with the default parameters [45]. Docking of compounds to the human RANKL dimer was carried out using AutoDock VINA v.1.1.2 [38] with the exhaustiveness level set to 10. The ranking of compounds for each library was based on the Vina score and visual investigation of the docked poses was performed using VMD v.1.9.3 [46]. Rendering of the figures was done using the open-source variant of PyMol v.1.8.4. Processing of the chemical databases and calculation of chemical properties from SMILES was performed using the open-source program DataWarrior (OpenMolecules.org, https://openmolecules.org/datawarrior, accessed on 21 June 2023) [47].

### 4.2. Expression, Purification and Electrophoresis of Human RANKL

The extracellular domain of RANKL (Lys159–Asp317) was expressed in *E*. *coli* as a glutathione S-transferase (GST)-fusion protein, as previously described [48]. GST-RANKL was purified after capturing on glutathione beads, while soluble RANKL was eluted from its GST fusion partner by proteolytic cleavage with the type-14 human rhinovirus 3C protease (America Pharmacia Biotech). The concentration of protein in the samples was determined by the Bradford method using bovine albumin as standard. Proteins were separated by electrophoresis in 12% (*w*/*v*) SDS polyacrylamide gel electrophoresis (SDS-PAGE) [48].

### 4.3. Cell Culture of RAW264.7

The murine monocyte/macrophage cell line RAW264.7 (purchased from ATCC, Manassas, VA, USA) was cultured in DMEM (Gibco BRL, Waltham, MA, USA) containing 10% heat-inactivated FBS. The cells were grown at 37 °C in a humid atmosphere containing 5% CO_2_.

### 4.4. RANKL-Induced Osteoclast Differentiation

Bone marrow (BM) cells were collected after flushing out of mouse femurs and tibiae, subjected to gradient purification using Ficoll-Paque (GE Healthcare, Chicago, IL, USA), plated in 96-well plates at a density of 6 × 10^4^ cells/well and cultured in alpha Minimum Essential Medium (aMEM) (Gibco) containing 10% fetal bovine serum supplemented with 40 ng/mL human RANKL, prepared as previously described, and 25 ng/mL macrophage colony stimulating factor (M-CSF) (R&D Systems, Minneapolis, MN, USA) for 5 days. The RAW264.7 cells were seeded at a density of 4 × 10^3^ cells/well and stimulated with 40 ng/mL human RANKL for 4 days. All tested compounds were pre-incubated with RANKL at various concentrations in aMEM medium for 1 h at room temperature and then added to cell cultures that were replenished with fresh medium every 2 days. Osteoclasts were stained for tartrate-resistant acid phosphatase (TRAP) activity using a leukocyte acid phosphatase (TRAP kit) (Sigma-Aldrich, St. Louis, MO, USA). 

### 4.5. Quantitative TRAP Activity Assay

In the TRAP activity assay, BM cells or RAW264.7 cells were plated in 96-well plates at densities of 6 × 10^4^ cells/well or 4 × 10^3^ cells/well, respectively. BM cells were cultured in aMEM medium containing 10% fetal bovine serum supplemented with 40 ng/mL RANKL and 25 ng/mL M-CSF, whereas RAW264.7 cells were stimulated only with RANKL (R&D Systems) for 4 days. Then, cells were lysed in an ice-cold phosphate buffer containing 0.1% Triton X-100. Lysates were added to 96-well plates containing phosphatase substrate (p-nitrophenol phosphate, Sigma-Aldrich) and 40 mM tartrate acid buffer and incubated at 37 °C for 30 min. The reaction was stopped with the addition of 0.5 N NaOH. Absorbance was measured at 405 nm on a microplate reader (Optimax, Molecular Devices, Silicon Valley, CA, USA). TRAP activity was normalized to total protein, which was determined using the Bradford assay (Bio-Rad, Hercules, CA, USA). Percentages of TRAP activity were calculated relatively to the absorbance of the positive control (untreated). IC_50_ values (mean ± standard error of the mean calculated from five or more measuring points) were determined from three independent experiments.

### 4.6. Viability Assay

Cell viability was evaluated in preosteoclasts (bone-marrow-derived macrophages, BMMs) using the MTT assay as previously described [29]. Briefly, cells were seeded at a density of 10^5^ cells/well in 96-well plates and incubated with all tested compounds for 48 h in aMEM containing 10% fetal bovine serum supplemented with 25 ng/mL M-CSF (R&D Systems). After removal of the medium, each well was incubated with 0.5 mg/mL MTT (Sigma-Aldrich) in aMEM serum-free medium at 37 °C for 2 h. Upon removal of the medium, 200 μL of DMSO was added and the absorbance was measured at 550 nm on a microplate reader (Optimax, Molecular Devices). LC_50_ values (mean ± SD calculated from five or more measuring points) were determined from three independent experiments.

### 4.7. General Procedure for the Preparation of Compounds PRAN-1.1 to PRAN-3.10

Cu(I)-catalyzed coupling of aryl bromide **2** with 5-nitro-1*H*-indole (**1**, Figure 5) was performed by mixing **1** (1.0 equiv.) and **2** (4.0 equiv.) with Cu(I) (1.0 equiv.) and Cs_2_CO_3_ (1.4 equiv.) in 1.0 mL dry dimethylformamide (DMF) and stirred at 164 °C under inert (Ar) atmosphere until full conversion, typically within 4 h. After cooling at room temperature (r.t.) the mixture was dissolved in dichloromethane (DCM), filtered through Celite and washed with H_2_O (3 × 10 mL). The organic phase was then dried over sodium sulfate (Na_2_SO_4_) and condensed under reduced pressure. The resulting residue was further purified with flash column chromatography to yield the final products (**3**, Figure 5).

Platinum-catalyzed reduction of the nitro-product **3** to the corresponding amine **4** (Figure 5) was carried out by dissolving 1.0 mmol of **3** in a 5.0 mL absolute ethanol (EtOH), and the solution was carefully degassed under inert atmosphere before addition of 10% platinum on activated carbon (Pt/C, cat. 10–20 mg). Hydrogenation was performed with H_2_ at 1 atm under r.t. for 17 h. The mixture was then filtered through Celite and condensed under reduced pressure. The resulting residue was further purified with flash column chromatography to yield the corresponding amine **4.**

Coupling of each amine **4** with the desired sulfonyl chloride **5** (Figure 5) was performed by mixing **4** (1.0 equiv.) with **5** (1.3 equiv.) in dry acetonitrile (CH_3_CN) under Ar atmosphere, and then pyridine (1.6 equiv.) was added under r.t. After stirring for 18 h, ethyl acetate (20 mL) was added and the organic phase was washed with 0.1 N HCl (2 × 10 mL), saturated NaHCO_3_ (2 × 10 mL) and saturated NaCl (3 × 5 mL). The organic phase was collected and dried over Na_2_SO_4_ and condensed under reduced pressure. The resulting residue was purified with flash column chromatography to obtain the final product **6** (Figure 5).

### 4.8. Characterization and Purification Methods

NMR spectra of the compounds were recorded on a Bruker Advance spectrometer operating at 500 MHz for proton (^1^H NMR) and 126 MHz for carbon (^13^C NMR); chemical shifts were reported in ppm (δ) relative to residual protons in deuterated solvent peaks (Appendix A). All final compounds reported within the Appendix A were purified to ≥95% as determined by liquid chromatography–mass spectrometry (LCMS). Data were acquired on a Shimadzu LCMS system equipped with a DGU-20A3 degasser, an LC-20AD binary gradient pump, an SPD-20A photodiode array detector, an SIL-20AC autosampler, a CTO-20AC column oven, an LCMS-2010EV single quadrupole mass spectrometer and a Purospher RP8 250 × 4.6 mm × 5.0 μm column. Detection wavelengths were set at 216 nm and 264 nm, mainly using mobile phases of H_2_O with 0.9% acetic acid (A) and acetonitrile (B). The gradient profile for each compound was reported with the LCMS spectra in the Appendix A, and purity was reported as the % area of the highest peak (Appendix A).

## Data Availability

The data presented in this study are available in Appendix A and can be provided on request from the corresponding authors.

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
