# Peer review of "Structure-Based Discovery of Receptor Activator of Nuclear Factor-κB Ligand (RANKL)-Induced Osteoclastogenesis Inhibitors"

_ijms, 2023, doi:10.3390/ijms241411290_

Round 1

Reviewer 1 Report

In this article the authors have presented some inhibitor molecules of RANKL-induced osteoclastogenesis by screening of libraries of commercially available small molecules. They have also synthesized some derivatives of the lead molecules and evaluated them in terms of inhibitory effect and cellular toxicity level. The research design is good, experiments are adequately described, and results clearly presented. However, there are some items that can be clarified:

1.       In line 162, the authors made a generalized statement on the properties of the inhibitor molecule and chose not to use a filtering criterion regarding that, however, the inhibitor molecules of different proteins/enzymes can come in a wide array of molecular weight, hydrophobicity, and solubility. Hence that filtering criteria could have been useful.

2.       The identified inhibitor molecules do have a diverse range of chemical backbone structure. The authors did not present any rationale to explain this diversity. Usually for a particular protein the structure of the inhibitors follows a common structural backbone with some variations in the side chains or functional groups.

3.       The Structure-Activity-relationship needs to be more robust, describing what elements of the structural backbone is responsible for showing the inhibitory effect and what are responsible for the toxicity. Authors only provided explanation about functional group interaction.

4.       The authors have modeled their inhibitors but did not give any inhibitor bound x-ray crystal structure of the protein.

5.       4.The authors did not provide any information on why some compounds are toxic, i.e. which pathway they are blocking.

English is mostly good.

Reviewer 3 Report

Title: OK

Abstract: OK

Keywords: It is important to present some keywords

 Introduction: The introduction is well structured, the contextualization of the topic is clear, concise and very well supported. The knowledge gap is well defined, finally the authors induce the reader to continue reading their research.

Results and Discussion:  The results are compelling, they are well organized and the discussion is very well supported. It would be relevant to report the solubility scale used, since in strict terms no compound is insoluble. In addition, it would be important to explain why solubility in DMSO is important, and not in compounds such as water or octanol, which are very relevant data in computational studies.

Conclusions: In the conclusions section, the authors report results. What are the proposals of the work, the results are outstanding, the quality of the document is magnificent, so the Conclusions section deserves to be substantially improved.

Methods and Materials: The section is clear and very well structured, which guarantees the quality of the results, as well as guaranteeing that other researchers can replicate this research.
